# Base-Catalyzed Nucleophilic Addition Reaction of Indoles with Vinylene Carbonate: An Approach to Synthesize 4-Indolyl-1,3-dioxolanones

**DOI:** 10.3390/molecules28217450

**Published:** 2023-11-06

**Authors:** Xia Chen, Xiao-Yu Zhou, Ming Bao

**Affiliations:** 1School of Chemistry and Materials Engineering, Liupanshui Normal University, Liupanshui 553004, China; xia811@live.cn; 2State Key Laboratory of Fine Chemicals, Dalian University of Technology, Dalian 116024, China

**Keywords:** nucleophilic addition, *N*-functionalization, N-H indoles, vinylene carbonate, 4-indolyl-1,3-dioxolanones

## Abstract

The *N*-functionalized indole is a privileged structural framework in a wide range of bioactive molecules. The nucleophilic addition between indoles with vinylene carbonate proceeded smoothly in the presence of K_2_CO_3_ as the catalyst to produce novel indolyl-containing skeletons and 4-indolyl-1,3-dioxolanones in satisfactory to excellent yields (up to >97% yield). Various synthetically useful functional groups, such as halogen atoms, cyano, nitro, and methoxycarbonyl groups, remained intact during the regioselective *N*-H addition reactions. The developed catalytic system also could accommodate 2-naphthalenol to achieve the target O-H additive product in good yield.

## 1. Introduction

The indole moiety is a ubiquitous structural framework in natural products and widely recognized as privileged components in biologically, physiologically, and pharmacologically relevant compounds [1,2,3,4,5,6,7,8]. Therefore, an impressive number of practical techniques as well as methods have been developed for the synthesis of indole-containing compounds and related functionalization. Among them, *N*-functionalization of indoles is of particular importance for the synthesis of *N*-substituted indoles with bioactivity. The indoles normally serve as nucleophilic coupling partners to react with diverse electrophiles, such as activated olefins, ketones, imines, and alkynes [9,10,11,12,13]. Olefins, as a stable and atom-economic synthon, are one of the fundamental synthetic materials in organic chemistry and provide a desired alternative for alkylating agents. As a result, extensive efforts have been devoted to realizing the Friedel–Crafts reaction of indoles with electron-deficient olefins for the preparation of C3-alkylated indoles [14,15,16,17,18]; in contrast, *N*-alkylation of indoles with olefins is much more challenging, due to the higher nucleophilicity toward the C3 position of indoles compared to the N1 site (Figure 1, Equation (1)). Typically, three main approaches have been developed to enhance the reactivity of the N1 position of indoles (Figure 1, Equation (2)). One approach involves the installation of a substituent at the C3 position to absolutely overcome the competitive transformations and thus create more opportunities for the *N*-functionalization [19,20]. Another approach involving the introduction of an electron-withdrawing group at the C2 position can increase the acidity of the *N*-H bond to dramatically avoid the competitive transformations from the nucleophilicity of the C3 position [21,22,23,24,25]. On the other hand, the N1-H bond possessing weak acidity [26] enables the generation of nitrogen anions in the presence of a base, thus enhancing the nucleophilicity of N1. Indeed, such an elegant strategy is the most straightforward, economical, and convenient method to realize the *N*-functionalization of *N*-H indoles without complex transition-metal catalysts and organocatalysts.

Vinylene carbonate has been regarded as a valuable and versatile C2 surrogate in organic synthesis [27,28,29,30,31]. In particular, vinylene carbonate has been extensively employed in the transition-metal-catalyzed C−H bond activation and annulation, resulting in the construction of value-added *N*-heterocyclic frameworks [32,33,34,35,36]. Vinylene carbonate is an inexpensive electron-deficient olefin; nevertheless, the nucleophilic additions between indoles with vinylene carbonate for the preparation of indolyl-containing ethylene carbonates have not been reported. Furthermore, indolyl-containing ethylene carbonates can undergo further transformation so as to obtain structurally diverse indolyl-containing skeletons [37,38,39,40,41]. 

In the course of our continuous research on the development of novel methods to achieve indole functionalization [42,43,44,45,46,47,48,49,50,51], we achieved the efficient synthesis of 4-indolyl-1,3-dioxolanones through the base-catalyzed nucleophilic addition reaction of *N*-H indoles with vinylene carbonate. The results are described in this paper.

## 2. Results

Initially, the nucleophilic addition reaction of indole (**1a**) with vinylene carbonate (**2**, 4 equiv.) was performed as a model to optimize the reaction parameters. The results are summarized in Table 1. The base catalyst was firstly screened using acetonitrile (CH_3_CN) as the solvent at 70 °C. Among the examined bases [triethylamine (NEt_3_), sodium bicarbonate (NaHCO_3_), sodium carbonate (Na_2_CO_3_), sodium acetate (NaOAc), potassium carbonate (K_2_CO_3_), sodium formate (HCOONa∙2H_2_O), DABCO (1,4-diaza[2.2.2]bicyclooctane), and DBU (1,8-diazabicyclo[5,4,0]-7-undecene)], K_2_CO_3_ proved to be the best base, providing the product, 4-(1*H*-indol-1-yl)-1,3-dioxolan-2-one (**3a**), with 71% yield (entries 1–8). The structure of product **3a** was confirmed through X-ray crystallographic analysis (Figure 1, CCDC 2299714). A relatively high yield was observed when the loading of catalyst K_2_CO_3_ was increased to 30 mol% and 40 mol%, respectively (entries 9 and 10 vs. entry 5). However, the yield was found to be decreased when the base loading was further increased (entry 11 vs. entries 9 and 10). The reaction temperature was subsequently screened using 40 mol% of K_2_CO_3_ as the catalyst. The results obtained indicated that 60 °C was the best reaction temperature (entry 10 vs. entries 12–14). Then, the solvent was investigated under 60 °C conditions. Non-polar solvents [benzene and toluene] and polar solvents [CH_3_CN, *tert*-butyl methyl ether (MTBE), 1,2-ethanediol dimethyl ether (DME), tetrahydrofuran (THF), and 1,4-dioxane] were examined, respectively (entry 13, 15–20). Except for CH_3_CN, the use of other solvents showed almost no reaction, and most of the starting materials were recovered. This reaction had a strong dependence on solvent, whereas the exact factors involved remain unclear. Relatively low yields were obtained when either decreasing or increasing the amounts of vinylene carbonate (entry 13 vs. entries 21 and 22). Therefore, the subsequent nucleophilic addition reaction of various *N*-H indoles with vinylene carbonate (4.0 equiv.) was performed in the presence of K_2_CO_3_ (40 mol%) in CH_3_CN (3 mL) at 60 °C.

Given the optimized reaction conditions, the scope and limitation of this reaction were explored, and the results are summarized in Figure 2. The electron-donating group, such as methyl (Me) and phenyl (Ph) linked on the pyrrole ring in substrate **1,** would severely hamper the reaction, and no target products were detected. Nevertheless, the electron-withdrawing group, such as methoxycarbonyl (COOMe) linked on the pyrrole ring, was tolerated well, and the corresponding addition product **3e** was obtained with 66% yield. The results suggested that the electron-donating group linked on the pyrrole ring could greatly weaken the acidity of the N–H bond, which was consistent with the formation of the indole nitrogen anion. In contrast, reactions of indole substrates **1** bearing the Me group linked on different positions of benzene ring proceeded smoothly under standard conditions; the corresponding 4-indolyl-1,3-dioxolanone products **3f**, **3n,** and **3w** were obtained in moderate yields. Other electron-donating groups, such as methoxyl (OMe) and benzyloxyl (OBn), were also tolerated well in this nucleophilic addition reaction, providing the desired products **3g**–**3h**, **3o**–**3p**, and **3x**–**3y** in 28–76% yields, respectively. Reactions of *N*-H indole substrates **1** bearing the electron-withdrawing group, such as COOMe, nitro (NO_2_), and cyano (CN), linked on different positions of the benzene ring were subsequently investigated under the optimized reaction conditions. The 4-indolyl-1,3-dioxolanone products **3k**–**3m**, **3t**–**3v**, and **3z** were obtained in good to excellent yields (82–97%). These results mentioned above indicated that the electron properties (electron donating or electron withdrawing) of substituents linked on either pyrrole or benzene ring significantly affect the reactivity of **1**. The reason for these results may be that the electron-withdrawing group, compared to electron-donating group, is more conducive to the formation and stabilization of indole nitrogen anion. The suitability of indole substrates **1i**–**1j**, **1q–1s,** and **1aa** having halogen atoms (Cl, Br, and I) in the current nucleophilic addition reaction was investigated. The desired products **3i**–**3j**, **3q–3s,** and **3aa** were also obtained in good to excellent yields (80–92%). The survival of synthetically useful functional groups, such as halogen atoms (Cl, Br, and I), CN, and CO_2_Me, under the reaction conditions will further increase structural diversification. In general, this nucleophilic addition exhibits broad scope and proceeds efficiently with electron-poor and -rich indoles, especially electron-poor indoles. However, the presence of groups at the C2-position of indole, with significant steric hindrance, would severely hamper the reaction.

Continued substrate extension studies showed that the 2-naphthalenol (**4**) was also applicable to this transformation and provided the desired addition product, 4-(naphthalen-2-yloxy)-1,3-dioxolan-2-one (**5**), in 87% yield (Figure 3, Equation (1)). Subsequently, instead of the benzene ring in substrate **1** studies addressed a pyridine ring and found that the reactions of 1*H*-pyrrolo[3,2-*b*]pyridine (**6**) and 1*H*-pyrrolo[2,3-*b*]pyridine (**8**) also proceeded smoothly to furnish the addition products **7** and **9** in 81% and 88% yields, respectively (Figure 3, Equations (2) and (3)).

Based on the experimental results and previous reports [48,49,52], a plausible reaction mechanism for the synthesis of *N*-functionalized indoles is depicted in Figure 4. Initially, indole nitrogen anion **10** was generated in situ through the deprotonation of indole in the presence of the base K_2_CO_3_. The indole nitrogen anion **10** then underwent nucleophilic addition to vinylene carbonate. Meanwhile, KHCO_3_ provided a proton to yield the addition product **3** and the regenerated base K_2_CO_3_.

## 3. Materials and Methods

### 3.1. General Information

Unless otherwise noted, all reactions were carried out in oven-dried 25-mL Schlenk tubes under a nitrogen atmosphere. An IKA plate was used as the heat source. All reagents and solvents were of pure analytical grade. Thin layer chromatography (TLC) was performed on HSGF254 silica gel, pre-coated on glass-backed plates coated with 0.2 mm silica and was revealed with either a UV lamp (λ_max_ = 254 nm). The products were purified by using flash column chromatography on silica gel 200–300 mesh. ^1^H and ^13^C NMR spectra were recorded on a 400 MHz spectrometer (^1^H 400 MHz, ^13^C 101 MHz) using *d*_6_-DMSO or CDCl_3_ as the solvent with tetramethylsilane (TMS) as the internal standard at room temperature. The chemical shifts are reported in ppm downfield (*δ*) from TMS, and the coupling constants *J* are given in Hz. The peak patterns are indicated as follows: s, singlet; d, doublet; t, triplet; q, quartet; m, multiplet. The NMR spectra of all compounds are demonstrated in Appendix A. High-resolution mass spectra were recorded on either Q-TOF mass spectrometry or LTQ Orbitrap XL mass spectrometry. X-ray crystallography analysis was performed on a Bruker D8 Quest X-ray diffractionmeter.

### 3.2. Synthetic Procedures

#### 3.2.1. The Typical Procedure for the Synthesis of 4-Indolyl-1,3-dioxolanones 3

A mixture of indoles **1** (0.50 mmol), vinylene carbonate (172 mg, 2.0 mmol, 4.0 equiv), and K_2_CO_3_ (27.6 mg, 0.20 mmol, 40 mol%) in CH_3_CN (3 mL) was added into a Schlenk flask (25 mL) and stirred at 60 °C. After the reaction was finished, the solvent was evaporated under reduced pressure and the residue was purified by using column chromatography (petroleum ether/ethyl acetate 5:1 to 1:1) to provide the product **3**.

4-(1*H*-indol-1-yl)-1,3-dioxolan-2-one (**3a**): Yield: 80%, 80.9 mg, white solid, mp 132–134 °C, R_f_ = 0.41 (H/E = 2:1). ^1^H NMR (400 MHz, *d*_6_-DMSO) *δ* 7.71 (d, *J* = 3.4 Hz, 1H), 7.64 (d, *J* = 7.8 Hz, 1H), 7.59 (d, *J* = 8.2 Hz, 1H), 7.28 (t, *J* = 7.7 Hz, 1H), 7.23 (t, *J* = 6.3 Hz, 1H), 7.18 (t, *J* = 7.5 Hz, 1H), 6.69 (d, *J* = 3.3 Hz, 1H), 5.04 (d, *J* = 6.3 Hz, 2H). ^13^C NMR (101 MHz, *d*_6_-DMSO) *δ* 154.1, 136.0, 129.5, 126.2, 123.2, 121.6, 121.6, 110.4, 105.6, 82.3, 68.4. [M + H]^+^ calculated for C_11_H_10_NO_3_, 204.0661; found 204.0657.methyl 1-(2-oxo-1,3-dioxolan-4-yl)-1*H*-indole-3-carboxylate (**3e**): Yield: 66%, 86.6 mg, white solid, mp 206–208 °C, R_f_ = 0.32 (H/E = 2:1). ^1^H NMR (400 MHz, *d*_6_-DMSO) *δ* 8.53 (s, 1H), 8.10 (d, *J* = 7.8 Hz, 1H), 7.63 (d, *J* = 8.1 Hz, 1H), 7.42–7.33 (m, 2H), 7.28–7.21 (m, 1H), 5.17–4.99 (m, 2H), 3.86 (s, 3H). ^13^C NMR (101 MHz, *d*_6_-DMSO) *δ* 164.5, 153.8, 136.0, 133.5, 126.9, 124.4, 123.5, 121.7, 111.2, 109.5, 82.4, 68.3, 51.6. HRMS (ESI) *m*/*z*: [M + H]^+^ calculated for C_13_H_12_NO_5_, 262.0715; found 262.0714.4-(4-methyl-1*H*-indol-1-yl)-1,3-dioxolan-2-one (**3f**): Yield: 53%, 57.7 mg, white solid, mp 118–120 °C, R_f_ = 0.36 (H/E = 2:1). ^1^H NMR (400 MHz, *d*_6_-DMSO) *δ* 7.68 (d, *J* = 3.4 Hz, 1H), 7.39 (d, *J* = 8.2 Hz, 1H), 7.23–7.14 (m, 2H), 6.98 (d, *J* = 7.2 Hz, 1H), 6.72 (d, *J* = 3.3 Hz, 1H), 5.03 (d, *J* = 6.4 Hz, 2H), 2.49 (s, 3H). ^13^C NMR (101 MHz, *d*_6_-DMSO) *δ* 154.1, 135.7, 130.6, 129.4, 125.6, 123.3, 121.7, 107.9, 104.1, 82.5, 68.4, 18.7. HRMS (ESI) *m*/*z*: [M + H]^+^ calculated for C_12_H_12_NO_3_, 218.0817; found 218.0815.4-(4-methoxy-1*H*-indol-1-yl)-1,3-dioxolan-2-one (**3g**): Yield: 74%, 86.0 mg, white solid, mp 172–174 °C, R_f_ = 0.33 (H/E = 2:1). ^1^H NMR (400 MHz, *d*_6_-DMSO) *δ* 7.59 (d, *J* = 3.4 Hz, 1H), 7.23–7.15 (m, 3H), 6.69 (d, *J* = 7.6 Hz, 1H), 6.66 (d, *J* = 3.2 Hz, 1H), 5.01 (d, *J* = 6.4 Hz, 2H), 3.89 (s, 3H). ^13^C NMR (101 MHz, *d*_6_-DMSO) *δ* 154.0, 153.4, 137.3, 124.7, 124.4, 119.7, 103.5, 102.6, 101.9, 82.5, 68.4, 55.6. HRMS (ESI) *m*/*z*: [M + H]^+^ calculated for C_12_H_12_NO_4_, 234.0766; found 234.0765.4-(4-(benzyloxy)-1*H*-indol-1-yl)-1,3-dioxolan-2-one (**3h**): Yield: 65%, 100.3 mg, white solid, mp 146–148 °C, R_f_ = 0.32 (H/E = 2:1). ^1^H NMR (400 MHz, *d*_6_-DMSO) *δ* 7.61 (d, *J* = 3.4 Hz, 1H), 7.51 (d, *J* = 7.6 Hz, 2H), 7.41 (t, *J* = 7.4 Hz, 2H), 7.34 (t, *J* = 7.1 Hz, 1H), 7.21–7.14 (m, 3H), 6.78 (d, *J* = 6.5 Hz, 1H), 6.71 (d, *J* = 3.3 Hz, 1H), 5.26 (s, 2H), 5.02 (d, *J* = 6.4 Hz, 2H). ^13^C NMR (101 MHz, *d*_6_-DMSO) *δ* 154.0, 152.4, 137.8, 137.4, 128.9, 128.2, 127.9, 124.9, 124.3, 120.1, 103.7, 103.4, 102.6, 82.5, 69.6, 68.4. HRMS (ESI) *m*/*z*: [M + H]^+^ calculated for C_18_H_16_NO_4_, 310.1079; found 310.1075.4-(4-chloro-1*H*-indol-1-yl)-1,3-dioxolan-2-one (**3i**): Yield: 84%, 100.0 mg, white solid, mp 130–132 °C, R_f_ = 0.48 (H/E = 2:1). ^1^H NMR (400 MHz, *d*_6_-DMSO) *δ* 7.86 (d, *J* = 3.5 Hz, 1H), 7.59 (d, *J* = 7.6 Hz, 1H), 7.33–7.22 (m, 3H), 6.73 (d, *J* = 3.3 Hz, 1H), 5.05 (d, *J* = 6.3 Hz, 2H). ^13^C NMR (101 MHz, *d*_6_-DMSO) *δ* 153.9, 136.8, 127.8, 127.5, 125.5, 124.2, 121.2, 109.7, 103.4, 82.3, 68.5. HRMS (ESI) *m*/*z*: [M + H]^+^ calculated for C_11_H_9_ClNO_3_, 238.0271; found 238.0266.4-(4-bromo-1*H*-indol-1-yl)-1,3-dioxolan-2-one (**3j**): Yield: 87%, 123.0 mg, white solid, mp 154–156 °C, R_f_ = 0.36 (H/E = 2:1). ^1^H NMR (400 MHz, *d*_6_-DMSO) *δ* 7.87 (d, *J* = 3.4 Hz, 1H), 7.64 (d, *J* = 8.3 Hz, 1H), 7.41 (d, *J* = 7.6 Hz, 1H), 7.23 (t, *J* = 7.3 Hz, 2H), 6.65 (d, *J* = 3.4 Hz, 1H), 5.05 (d, *J* = 6.3 Hz, 2H). ^13^C NMR (101 MHz, *d*_6_-DMSO) *δ* 153.9, 136.4, 133.2, 129.7, 127.5, 124.5, 124.3, 114.5, 110.2, 105.1, 82.3, 68.5. HRMS (ESI) *m*/*z*: [M + H]^+^ calculated for C_11_H_9_BrNO_3_, 281.9766; found 281.9758.methyl 1-(2-oxo-1,3-dioxolan-4-yl)-1*H*-indole-4-carboxylate (**3k**): Yield: 91%, 119.0 mg, white solid, mp 146–148 °C, R_f_ = 0.39 (H/E = 2:1). ^1^H NMR (400 MHz, *d*_6_-DMSO) *δ* 7.92 (d, *J* = 7.5 Hz, 2H), 7.88 (d, *J* = 7.5 Hz, 1H), 7.41 (t, *J* = 7.9 Hz, 1H), 7.30 (t, *J* = 6.2 Hz, 1H), 7.19 (d, *J* = 3.3 Hz, 1H), 5.07 (dd, *J* = 8.6, 3.5 Hz, 2H), 3.92 (s, 3H). ^13^C NMR (101 MHz, *d*_6_-DMSO) *δ* 167.1, 153.9, 136.9, 128.8, 128.3, 124.5, 122.7, 121.8, 115.6, 106.2, 82.0, 68.5, 52.4. HRMS (ESI) *m*/*z*: [M + H]^+^ calculated for C_13_H_12_NO_5_, 262.0715; found 262.0708.4-(4-nitro-1*H*-indol-1-yl)-1,3-dioxolan-2-one (**3l**): Yield: 90%, 111.7 mg, light yellow solid, mp 188–190 °C, R_f_ = 0.30 (H/E = 2:1). ^1^H NMR (400 MHz, *d*_6_-DMSO) *δ* 8.20 (d, *J* = 8.0 Hz, 1H), 8.15 (d, *J* = 8.5 Hz, 2H), 7.53 (t, *J* = 8.1 Hz, 1H), 7.36 (t, *J* = 6.2 Hz, 1H), 7.26 (d, *J* = 3.3 Hz, 1H), 5.13–5.04 (m, 2H). ^13^C NMR (101 MHz, *d*_6_-DMSO) *δ* 153.8, 140.2, 138.2, 131.0, 123.1, 122.8, 119.0, 118.1, 104.8, 81.9, 68.67. HRMS (ESI) *m*/*z*: [M + H]^+^ calculated for C_11_H_9_N_2_O_5_, 249.0511; found 249.0510.1-(2-oxo-1,3-dioxolan-4-yl)-1*H*-indole-4-carbonitrile (**3m**): Yield: 91%, 112.2 mg, white solid, mp 166–168 °C, R_f_ = 0.35 (H/E = 2:1). ^1^H NMR (400 MHz, *d*_6_-DMSO) *δ* 8.05 (d, *J* = 3.4 Hz, 1H), 7.99 (d, *J* = 8.4 Hz, 1H), 7.72 (d, *J* = 7.4 Hz, 1H), 7.46 (t, *J* = 7.9 Hz, 1H), 7.30 (t, *J* = 6.3 Hz, 1H), 6.85 (d, *J* = 3.3 Hz, 1H), 5.06 (d, *J* = 6.3 Hz, 2H). ^13^C NMR (101 MHz, *d*_6_-DMSO) *δ* 153.8, 135.9, 130.4, 129.6, 126.9, 123.4, 118.4, 116.0, 103.5, 102.8, 82.0, 68.6. HRMS (ESI) *m*/*z*: [M + H]^+^ calculated for C_12_H_9_N_2_O_3_, 229.0613; found 229.0612.4-(5-methyl-1*H*-indol-1-yl)-1,3-dioxolan-2-one (**3n**): Yield: 51%, 55.6 mg, light yellow solid, mp 112–114 °C, R_f_ = 0.36 (H/E = 2:1). ^1^H NMR (400 MHz, *d*_6_-DMSO) *δ* 7.64 (d, *J* = 3.4 Hz, 1H), 7.47–7.40 (m, 2H), 7.18 (t, *J* = 6.3 Hz, 1H), 7.10 (d, *J* = 8.4 Hz, 1H), 6.58 (d, *J* = 3.3 Hz, 1H), 5.02 (d, *J* = 6.3 Hz, 2H), 2.39 (s, 3H). ^13^C NMR (101 MHz, *d*_6_-DMSO) *δ* 154.1, 134.3, 133.2, 130.3, 129.9, 126.3, 124.6, 121.2, 110.1, 105.1, 82.5, 68.3, 21.4. HRMS (ESI) *m*/*z*: [M + H]^+^ calculated for C_12_H_12_NO_3_, 218.0817; found 218.0810.4-(5-methoxy-1*H*-indol-1-yl)-1,3-dioxolan-2-one (**3o**): Yield: 28%, 33.0 mg, white solid, mp 141–143 °C, R_f_ = 0.32 (H/E = 2:1). ^1^H NMR (400 MHz, CDCl_3_) *δ* 7.32 (d, *J* = 8.6 Hz, 1H), 7.21 (d, *J* = 3.4 Hz, 1H), 7.15 (d, *J* = 1.9 Hz, 1H), 7.00 (dd, *J* = 8.9, 2.0 Hz, 1H), 6.73 (dd, *J* = 7.2, 5.1 Hz, 1H), 6.64 (d, *J* = 3.3 Hz, 1H), 4.98-4.85 (m, 2H), 3.91 (s, 3H). ^13^C NMR (101 MHz, CDCl_3_) *δ* 155.5, 153.3, 130.5, 130.2, 124.9, 113.4, 110.1, 106.2, 103.7, 82.2, 67.9, 55.8. HRMS (ESI) *m*/*z*: [M + H]^+^ calculated for C_12_H_12_NO_4_, 234.0766; found 234.0758.4-(5-(benzyloxy)-1*H*-indol-1-yl)-1,3-dioxolan-2-one (**3p**): Yield: 41%, 63.3 mg, white solid, mp 146–148 °C, R_f_ = 0.32 (H/E = 2:1). ^1^H NMR (400 MHz, *d*_6_-DMSO) δ 7.66 (d, *J* = 3.2 Hz, 1H), 7.47 (d, *J* = 8.1 Hz, 3H), 7.40 (t, *J* = 7.4 Hz, 2H), 7.33 (d, *J* = 6.9 Hz, 1H), 7.23 (s, 1H), 7.16 (t, *J* = 6.2 Hz, 1H), 7.00 (d, *J* = 8.9 Hz, 1H), 6.59 (d, *J* = 3.2 Hz, 1H), 5.13 (s, 2H), 5.01 (d, *J* = 6.3 Hz, 2H). ^13^C NMR (101 MHz, *d*_6_-DMSO) *δ* 154.10, 154.08, 137.9, 131.0, 130.2, 128.9, 128.2, 128.1, 126.9, 113.5, 111.1, 105.4, 105.0, 82.6, 70.1, 68.3. HRMS (ESI) *m*/*z*: [M + H]^+^ calculated for C_18_H_16_NO_4_, 310.1079; found 310.1078.4-(5-chloro-1*H*-indol-1-yl)-1,3-dioxolan-2-one (**3q**): Yield: 85%, 100.4 mg, white solid, mp 127–129 °C, R_f_ = 0.37 (H/E = 2:1). ^1^H NMR (400 MHz, *d*_6_-DMSO) *δ* 7.81–7.79 (m, 1H), 7.70 (s, 1H), 7.62 (d, *J* = 8.8 Hz, 1H), 7.30 (d, *J* = 8.7 Hz, 1H), 7.22 (t, *J* = 6.3 Hz, 1H), 6.68 (d, *J* = 3.4 Hz, 1H), 5.04 (d, *J* = 6.3 Hz, 2H). ^13^C NMR (101 MHz, *d*_6_-DMSO) *δ* 154.0, 134.6, 130.7, 127.8, 126.1, 123.1, 120.8, 112.0, 105.2, 82.2, 68.5. HRMS (ESI) *m*/*z*: [M + H]^+^ calculated for C_11_H_9_ClNO_3_, 238.0271; found 238.0265.4-(5-bromo-1*H*-indol-1-yl)-1,3-dioxolan-2-one (**3r**): Yield: 80%, 112.5 mg, white solid, mp 108–110 °C, R_f_ = 0.37 (H/E = 2:1). ^1^H NMR (400 MHz, *d*_6_-DMSO) *δ* 7.85 (s, 1H), 7.79 (d, *J* = 3.3 Hz, 1H), 7.58 (d, *J* = 8.7 Hz, 1H), 7.42 (d, *J* = 8.6 Hz, 1H), 7.22 (t, *J* = 6.3 Hz, 1H), 6.68 (d, *J* = 3.2 Hz, 1H), 5.03 (d, *J* = 6.3 Hz, 2H). ^13^C NMR (101 MHz, *d*_6_-DMSO) *δ* 153.9, 134.9, 131.4, 127.7, 125.7, 123.8, 114.1, 112.5, 105.1, 82.2, 68.4. HRMS (ESI) *m*/*z*: [M + H]^+^ calculated for C_11_H_9_BrNO_3_, 281.9766; found 281.9757.4-(5-iodo-1*H*-indol-1-yl)-1,3-dioxolan-2-one (**3s**): Yield: 86%, 141.6 mg, white solid, mp 134–136 °C, R_f_ = 0.36 (H/E = 2:1). ^1^H NMR (400 MHz, *d*_6_-DMSO) *δ* 8.02 (s, 1H), 7.73 (d, *J* = 3.3 Hz, 1H), 7.55 (d, *J* = 8.6 Hz, 1H), 7.45 (d, *J* = 8.7 Hz, 1H), 7.20 (t, *J* = 6.3 Hz, 1H), 6.65 (d, *J* = 3.2 Hz, 1H), 5.02 (d, *J* = 6.3 Hz, 2H). ^13^C NMR (101 MHz, *d*_6_-DMSO) *δ* 153.9, 135.3, 132.1, 131.1, 130.0, 127.2, 112.9, 104.8, 85.6, 82.1, 68.4. HRMS (ESI) *m*/*z*: [M + H]^+^ calculated for C_11_H_9_INO_3_, 329.9627; found 329.9619.methyl 1-(2-oxo-1,3-dioxolan-4-yl)-1*H*-indole-5-carboxylate (**3t**): Yield: 82%, 107.2 mg, white solid, mp 164–166 °C, R_f_ = 0.34 (H/E = 2:1). ^1^H NMR (400 MHz, *d*_6_-DMSO) *δ* 8.33 (s, 1H), 7.90 (d, *J* = 8.7 Hz, 1H), 7.86 (d, *J* = 3.3 Hz, 1H), 7.71 (d, *J* = 8.7 Hz, 1H), 7.28 (t, *J* = 6.3 Hz, 1H), 6.85 (d, *J* = 3.3 Hz, 1H), 5.05 (d, *J* = 6.2 Hz, 2H), 3.87 (s, 3H). ^13^C NMR (101 MHz, *d*_6_-DMSO) *δ* 167.2, 153.9, 138.7, 129.2, 127.7, 124.0, 123.8, 123.1, 110.6, 106.7, 82.1, 68.6, 52.4. HRMS (ESI) *m*/*z*: [M + H]^+^ calculated for C_13_H_12_NO_5_, 262.0715; found 262.0708.4-(5-nitro-1*H*-indol-1-yl)-1,3-dioxolan-2-one (**3u**): Yield: 91%, 112.5 mg, light yellow solid, mp 188–190 °C, R_f_ = 0.40 (H/E = 1:1). ^1^H NMR (400 MHz, *d*_6_-DMSO) *δ* 8.64 (s, 1H), 8.18 (d, *J* = 9.1 Hz, 1H), 8.00 (d, *J* = 3.4 Hz, 1H), 7.83 (d, *J* = 9.1 Hz, 1H), 7.32 (t, *J* = 6.3 Hz, 1H), 6.97 (d, *J* = 3.4 Hz, 1H), 5.12–4.99 (m, 2H). ^13^C NMR (101 MHz, *d*_6_-DMSO) *δ* 153.8, 142.6, 139.2, 129.5, 128.9, 118.4, 118.3, 111.2, 107.6, 81.9, 68.7. HRMS (ESI) *m*/*z*: [M + H]^+^ calculated for C_11_H_9_N_2_O_5_, 249.0511; found 249.0503.1-(2-oxo-1,3-dioxolan-4-yl)-1*H*-indole-5-carbonitrile (**3v**): Yield: 97%, 111.0 mg, white solid, mp 186–188 °C, R_f_ = 0.40 (H/E = 1:1). ^1^H NMR (400 MHz, *d*_6_-DMSO) *δ* 8.19 (s, 1H), 7.95 (d, *J* = 3.4 Hz, 1H), 7.81 (d, *J* = 8.6 Hz, 1H), 7.67 (d, *J* = 8.6 Hz, 1H), 7.30 (t, *J* = 6.2 Hz, 1H), 6.83 (d, *J* = 3.3 Hz, 1H), 5.05 (d, *J* = 5.8 Hz, 2H). ^13^C NMR (101 MHz, *d*_6_-DMSO) *δ* 153.8, 137.9, 129.3, 128.6, 127.0, 126.1, 120.5, 111.9, 106.2, 103.9, 81.9, 68.6. HRMS (ESI) *m*/*z*: [M + H]^+^ calculated for C_12_H_9_N_2_O_3_, 229.0613; found 229.0607.**4-(6-methyl-1*H*-indol-1-yl)-1,3-dioxolan-2-one (3w):** Yield: 47%, 51.2 mg, white solid, mp 130–132 °C, R_f_ = 0.36 (H/E = 2:1). ^1^H NMR (400 MHz, *d*_6_-DMSO) *δ* 7.61 (d, *J* = 3.4 Hz, 1H), 7.51 (d, *J* = 8.0 Hz, 1H), 7.38 (s, 1H), 7.18 (t, *J* = 6.3 Hz, 1H), 7.01 (d, *J* = 8.1 Hz, 1H), 6.61 (d, *J* = 3.2 Hz, 1H), 5.02 (d, *J* = 6.3 Hz, 2H), 2.44 (s, 3H). ^13^C NMR (101 MHz, *d*_6_-DMSO) *δ* 154.1, 133.2, 132.5, 127.3, 125.4, 123.2, 121.2, 110.3, 105.5, 82.3, 68.3, 22.0. [M + H]^+^ calcd for C_12_H_12_NO_3_, 218.0817; found 218.0810.4-(6-methoxy-1*H*-indol-1-yl)-1,3-dioxolan-2-one (**3x**): Yield: 55%, 63.9 mg, white solid, mp 152–154 °C, R_f_ = 0.34 (H/E = 2:1). ^1^H NMR (400 MHz, *d*_6_-DMSO) *δ* 7.54 (d, *J* = 3.5 Hz, 1H), 7.49 (d, *J* = 8.6 Hz, 1H), 7.23 (t, *J* = 6.3 Hz, 1H), 7.19 (s, 1H), 6.82 (d, *J* = 8.6 Hz, 1H), 6.60 (d, *J* = 3.3 Hz, 1H), 5.02 (d, *J* = 6.3 Hz, 2H), 3.81 (s, 3H). ^13^C NMR (101 MHz, *d*_6_-DMSO) *δ* 157.0, 154.1, 137.2, 124.4, 123.2, 122.0, 111.2, 105.7, 94.4, 82.1, 68.4, 55.9. HRMS (ESI) *m*/*z*: [M + H]^+^ calculated for C_12_H_12_NO_4_, 234.0766; found 234.0765.4-(6-(benzyloxy)-1*H*-indol-1-yl)-1,3-dioxolan-2-one (**3y**): Yield: 55%, 85.0 mg, white solid, mp 139–141 °C, R_f_ = 0.36 (H/E = 2:1). ^1^H NMR (400 MHz, *d*_6_-DMSO) *δ* 7.55 (d, *J* = 3.4 Hz, 1H), 7.52–7.48 (m, 3H), 7.41 (t, *J* = 7.4 Hz, 2H), 7.37–7.31 (m, 2H), 7.21 (t, *J* = 6.3 Hz, 1H), 6.90 (d, *J* = 8.7 Hz, 1H), 6.60 (d, *J* = 3.3 Hz, 1H), 5.19-5.12 (m, 2H), 5.02 (d, *J* = 6.3 Hz, 2H). ^13^C NMR (101 MHz, *d*_6_-DMSO) *δ* 156.0, 154.1, 137.6, 137.2, 133.2, 128.9, 128.3, 128.2, 124.6, 123.5, 122.1, 111.7, 105.7, 95.8, 82.1, 70.2, 68.3. HRMS (ESI) *m*/*z*: [M + H]^+^ calculated for C_18_H_16_NO_4_, 310.1079; found 310.1071.methyl 1-(2-oxo-1,3-dioxolan-4-yl)-1*H*-indole-6-carboxylate (**3z**): Yield: 90%, 116.9 mg, white solid, mp 140–142 °C, R_f_ = 0.37 (H/E = 2:1). ^1^H NMR (400 MHz, *d*_6_-DMSO) *δ* 8.28 (s, 1H), 7.98 (d, *J* = 3.3 Hz, 1H), 7.80-7.73 (m, 2H), 7.38 (t, *J* = 6.3 Hz, 1H), 6.80 (d, *J* = 3.3 Hz, 1H), 5.16–4.92 (m, 2H), 3.89 (s, 3H). ^13^C NMR (101 MHz, *d*_6_-DMSO) *δ* 167.3, 153.9, 135.6, 133.2, 129.7, 124.3, 122.2, 121.5, 112.2, 105.87, 82.0, 68.5, 52.5. HRMS (ESI) *m*/*z*: [M + H]^+^ calculated for C_13_H_12_NO_5_, 262.0715; found 262.0712.4-(6-chloro-1*H*-indol-1-yl)-1,3-dioxolan-2-one (**3aa**): Yield: 92%, 109.0 mg, white solid, mp 152–154 °C, R_f_ = 0.38 (H/E = 2:1). ^1^H NMR (400 MHz, *d*_6_-DMSO) *δ* 7.76 (s, 2H), 7.64 (d, *J* = 8.4 Hz, 1H), 7.30–7.17 (m, 2H), 6.72 (d, *J* = 3.3 Hz, 1H), 5.03 (d, *J* = 6.0 Hz, 2H). ^13^C NMR (101 MHz, *d*_6_-DMSO) *δ* 153.9, 136.7, 133.2, 128.1, 128.0, 126.9, 122.9, 121.9, 110.6, 105.9, 82.0, 68.5. HRMS (ESI) *m*/*z*: [M + H]^+^ calculated for C_11_H_9_ClNO_3_, 238.0271; found 238.0264.

#### 3.2.2. The Typical Procedure for the Synthesis of **5**

To an oven-dried 25 mL Schlenk tube equipped with a magnetic stir bar was added 2-naphthol substrate **4** (0.50 mmol, 1.0 equiv.), vinylene carbonate (172 mg, 2.0 mmol, 4.0 equiv.), K_2_CO_3_ (27.6 mg, 0.20 mmol, 40 mol%), and CH_3_CN (3 mL) under an air atmosphere. The reaction mixture was stirred at 60 °C for 24 h and then cooled to room temperature. The solvent was removed under reduced pressure and the crude product was purified by using silica gel column chromatography to afford the desired products **5** as a white solid (100.0 mg, yield: 87%).

4-(naphthalen-2-yloxy)-1,3-dioxolan-2-one (**5**): Yield: 87%, 100.0 mg, white solid, mp 106–108 °C, R_f_ = 0.45 (H/E = 2:1). ^1^H NMR (400 MHz, *d*_6_-DMSO) *δ* 7.96 (d, *J* = 8.9 Hz, 1H), 7.92 (d, *J* = 8.2 Hz, 2H), 7.59 (s, 1H), 7.54 (t, *J* = 7.5 Hz, 1H), 7.46 (t, *J* = 7.5 Hz, 1H), 7.32 (dd, *J* = 8.9, 2.3 Hz, 1H), 6.71 (d, *J* = 3.9 Hz, 1H), 4.88 (dd, *J* = 9.9, 5.5 Hz, 1H), 4.66 (d, *J* = 10.0 Hz, 1H). ^13^C NMR (101 MHz, *d*_6_-DMSO) *δ* 154.1, 153.3, 134.1, 130.5, 130.2, 128.1, 127.7, 127.3, 125.4, 118.9, 111.3, 98.1, 70.9. HRMS (ESI) *m*/*z*: [M + H]^+^ calculated for C_13_H_11_O_4_, 231.0657; found 231.0654.

#### 3.2.3. The Typical Procedure for the Synthesis of **7** and **9**

To an oven-dried 25 mL Schlenk tube equipped with a magnetic stir bar was added pyrrolo-pyridine substrate **6** or **8** (0.50 mmol, 1.0 equiv.), vinylene carbonate (172 mg, 2.0 mmol, 4.0 equiv.), K_2_CO_3_ (27.6 mg, 0.20 mmol, 40 mol%), and CH_3_CN (3 mL) under an air atmosphere. The reaction mixture was stirred at 60 °C for 24 h, and then cooled to room temperature. The solvent was removed under reduced pressure and the crude product was purified by using silica gel column chromatography to afford the desired products **7** or **9**.

4-(1*H*-pyrrolo [3,2-b]pyridin-1-yl)-1,3-dioxolan-2-one (**7**): Yield: 81%, 82.2 mg, white solid, mp 176–178 °C, R_f_ = 0.31 (EtOAc). ^1^H NMR (400 MHz, *d*_6_-DMSO) *δ* 8.47 (d, *J* = 4.7 Hz, 1H), 8.05 (d, *J* = 3.5 Hz, 1H), 8.00 (d, *J* = 8.3 Hz, 1H), 7.29 (dd, *J* = 8.3, 4.6 Hz, 1H), 7.23 (t, *J* = 6.3 Hz, 1H), 6.81 (d, *J* = 3.4 Hz, 1H), 5.05 (d, *J* = 6.2 Hz, 2H). ^13^C NMR (101 MHz, *d*_6_-DMSO) *δ* 153.9, 147.6, 144.8, 129.8, 128.9, 118.0, 117.9, 106.1, 82.3, 68.4. HRMS (ESI) *m*/*z*: [M + H]^+^ calculated for C_10_H_9_N_2_O_3_, 205.0613; found 205.0613.4-(1*H*-pyrrolo [2,3-b]pyridin-1-yl)-1,3-dioxolan-2-one (**9**): Yield: 88%, 89.3 mg, white solid, mp 169-171 °C, R_f_ = 0.37 (H/E = 2:1). ^1^H NMR (400 MHz, *d*_6_-DMSO) *δ* 8.33 (d, *J* = 4.7 Hz, 1H), 8.07 (d, *J* = 7.8 Hz, 1H), 7.82 (d, *J* = 3.7 Hz, 1H), 7.28-7.12 (m, 2H), 6.66 (d, *J* = 3.7 Hz, 1H), 5.08-4.98 (m, 2H). ^13^C NMR (101 MHz, *d*_6_-DMSO) *δ* 154.3, 147.5, 143.8, 130.0, 128.0, 121.9, 118.0, 102.9, 81.6, 68.4. HRMS (ESI) *m*/*z*: [M + H]^+^ calculated for C_10_H_9_N_2_O_3_, 205.0613; found 205.0608.

### 3.3. X-ray Crystallographic Analysis

The structure of **3a** was determined based on single-crystal X-ray analysis. The detailed procedure was as follows: The **3a** solid was dissolved in AcOEt (1 mL). Then, the solvent was placed in the inner tube, and *n*-hexane (5 mL) was placed in the outer container. The crystals of **3a** were grown from solution, which is suitable for X-ray diffraction analysis.

CCDC No. 2299714 (**3a**) contains the supplementary crystallographic data for this paper. The crystal data can be obtained free of charge from the Cambridge Crystallographic Data Centre through www.ccdc.cam.ac.uk/datarequest/cif (accessed on 7 October 2023).

## 4. Conclusions

In summary, employing a base-catalyzed transition-metal-free strategy, we have successfully developed a convenient and alternative method to achieve *N*-functionalization of indoles with high regioselectivity. The nucleophilic addition reaction of various *N*-H indoles with vinylene carbonate proceeded smoothly to produce 4-indolyl-1,3-dioxolanones in satisfactory to excellent yields (up to >97% yield). These 4-Indolyl-1,3-dioxolanones, as novel indolyl-containing skeletons, are a type of indolyl-containing ethylene carbonate which could undergo further transformation to obtain structurally diverse bioactive molecules containing indole moiety. The readily available starting materials, the mild reaction conditions, and the experimental simplicity make the present methodology highly useful in the synthesis of 4-indolyl-1,3-dioxolanones. Based on the universality and significance of indole moiety in drugs, functional materials, and bioactive molecules, we believe that both this methodology and the synthesized indolyl-containing skeletons could be of interest to organic chemists and provide many medicinally active scaffolds.

## Data Availability

Data obtained in this project are contained within this article and are available upon request from the authors.

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
