# Peer review of "Base-Catalyzed Nucleophilic Addition Reaction of Indoles with Vinylene Carbonate: An Approach to Synthesize 4-Indolyl-1,3-dioxolanones"

_molecules, 2023, doi:10.3390/molecules28217450_

Round 1

Reviewer 1 Report

Comments and Suggestions for Authors

The manuscript "Base-Catalyzed Nucleophilic Addition Reaction of Indoles with Vinylene Carbonate: An Approach to Synthesize 4-Indolyl-1,3-dioxolanones" by X. Chen et al. dedicated to the development of a new efficient method for the synthesis of N-alkylindoles. The method comprises mild reaction conditions, cheap and available reagents and atom-economical transformation.

As a result, a series of 4-(indol-1-yl)-1,3-dioxolanols have been synthesized in moderate to excellent yields, scope of the reaction was studied and the method was extended to some other classes of related compounds such as pyrrolopyridines and even 2-naphthol. The plausible reaction mechanism was proposed based on experimental results and previous data. The article is compact, results are clearly presented, the cited references are mostly recent publication. The figures and schemes are appropriate and easy to understand.

To conclude, this article is suitable for publication in Molecules and would be of interest to wide journal readership, however, there are several small points to be addressed:

1. Page 4, line 127 and scheme 2. Despite the authors postulated that C7-substituted indoles would not readily undergo reaction with vinylene carbonate, there are no examples of such compounds on scheme 2. It seems reasonable either to add some 7-substituted indoles to the scope or to remove this phrase.

2. Page 10, Line 335. "o-phenylene diamines substrate" should be changed to "2-naphthol". The same is for supporting information file.

Author Response

1. Page 4, line 127. The phrase, C7-substituted indoles, has been removed. The change has been marked with color. 2. Page 10, Line 333. "o-phenylene diamines substrate" has been changed to "2-naphthol substrate ", and the same modification is for supporting information file. The change has been marked with color.

Reviewer 2 Report

Comments and Suggestions for Authors

”Base-Catalyzed Nucleophilic Addition Reaction of Indoles 2 with Vinylene Carbonate: An Approach to Synthesize 4-Indolyl-1,3-dioxolanone” submitted by Xia Chen, Xiao-Yu Zhou and Ming Chen.

The manuscript describes the discovery, optimization and scope and limitations of the N-alkylation of indoles using a large excess of vinylene carbonate and a catalytic amount of potassium carbonate to give 4-indolyl-1,3-dioxolane-2-ones. In addition to indoles, two azaindoles and 2-naphthol also participated in the reaction making N- and O-alkylated products. The yield of product ranged from 28% to 97%.  All in all, a significant amount of work is described. Overall, the text read OK and the products seems to be correctly identified and characterized (see below).  However, just putting a novel group on the nitrogen of an indole is fairly uninteresting. What is it good for, how can it be used in indole chemistry to make something important, is it important in medicinal chemistry, etc.?

Some issues:

P1, L 32-33: “regioselective N-alkylation of indoles remains difficult”  There are numerous papers published over the years describing N-alkylation of indoles with alkyl halides in the presence of a base or Michael additions so I’m not sure what the authors are trying to say.

P2: “This reaction mentioned above also represents an alternative way for indirect utilization of carbon dioxide (CO2) because vinylene carbonate is generally prepared 56 from ethylene carbonate (EC), and CO2 is chemical feedstock for EC production [37-40].” This statement should be removed from P2 and in the conclusion.

P2, L51: I don’t know what “metal-catalyzed C−H direction conversion” means.

The 1H NMR signals should be reported with more details. For example, for the parent compound, the set of resonances reported as 7.74-7.56 (m, 3H) is clearly three distinct sets of signals and should be reported as three different doublets with correct chemical shifts and coupling constants. The signals at 7.32-7.14 (m, 3H) and related cases throughout the manuscript should be treated in a similar fashion.

The reference selection is curious. It appears that the authors selected reference 1-6 randomly from the literature to show the importance of indoles.  It would be more appropriate to the select reviews that are summarizing this importance. References 7-9 are reviews but very specialized.

Comments on the Quality of English Language

There are some syntax and grammatical errors in the text but all in all good.

Author Response

  1. 4-Indolyl-1,3-dioxolanones, as novel indolyl-containing skeletons, are a type of indolyl-containing ethylene carbonates which could undergo further transformation to obtain structurally diverse bioactive molecules containing indole moiety. Furthermore, the inhibition zone test proved that compounds 4-(5-nitro-1H-indol-1-yl)-1,3-dioxolan-2-one and 4-chloro-5-(1H-indol-1-yl)-1,3-dioxolan-2-one had obvious inhibiting effects on escherichia coli and staphylococcus aureus. We will next focus on exploring the potential applications of this class of compounds in drug and function material discovery.
  2. P1, L 32-33: Most alkylation of indoles with olefins occur largely at the C3 position of indoles, owing to the innate nucleophilic nature of this position. In contrast, achieving N1-selectivity in the direct indole alkylation reactions is much more challenging, due to the low nucleophilicity of the N-H motif of indoles. Therefore, our original intention was to express that N-alkylation of indoles with olefins is more difficult compared to C3-alkylation of indoles. This expression may lead to confusion, so we have revised as “As a result, extensive efforts have been devoted to realizing Friedel-Crafts reaction of indoles with electron-deficient olefins for the preparation of C3-alkylated indoles [14–18]; in contrast, N-alkylation of indoles with olefins is much more challenging, due to the higher nucleophilicity toward the C3 position of indole compared to the N1 site (Scheme 1, eq. 1).” And the changes have been marked with color.
  3.  This statement and the relative references have been removed from P2, conclusion and references. The sequence number of references has also been modified accordingly.
  4.  P2, L51: It has been revised as “transition-metal-catalyzed C−H bond activation and annulation”, and marked with color.
  5.  For the parent compound (3a), the 1H NMR signals have been reported with more details [7.71 (d, J = 3.4 Hz, 1H), 7.64 (d, J = 7.8 Hz, 1H), 7.59 (d, J = 8.2 Hz, 1H), 7.28 (t, J = 7.7 Hz, 1H), 7.23 (t, J = 6.3 Hz, 1H), 7.18 (t, J = 7.5 Hz, 1H), 6.69 (d, J = 3.3 Hz, 1H), 5.04 (d, J = 6.3 Hz, 2H).], and the related cases such as 3k, 3l, 3t, 3x, 5 and 7 have been treated in a similar fashion. And the changes have been marked with color.
  6.  Appropriate reviews have been selected to show the importance of indoles instead of references 1-6, and the changes have been marked with color.
